# Moderate Drip Irrigation Level with Low Mepiquat Chloride Application Increases Cotton Lint Yield by Improving Leaf Photosynthetic Rate and Reproductive Organ Biomass Accumulation in Arid Region

**Hongyun Gao** [1,†], **Hui Ma** [1,†], **Aziz Khan** [1], **Jun Xia** [1], **Xianzhe Hao** [1], **Fangyong Wang** [2,*] **and Honghai Luo** [1,*]

1    Key Laboratory of Oasis Eco-Agriculture, Xinjiang Production and Construction Corps, Shihezi University, Shihezi, Xinjiang 832003, China; gaohongyun1995@163.com (H.G.); Mahui3219@163.com (H.M.); azizkhanturlandi@gmail.com (A.K.); xiajun9505@163.com (J.X.); haoxz386@163.com (X.H.)
2    Cotton Institute, Xinjiang Academy Agricultural and Reclamation Science, Shihezi, Xinjiang 832003, China
*    Correspondence: fangywang425@163.com (F.W.); luohonghai@shzu.edu.cn (H.L.);
     Tel.: +139-9953-8919 (F.W.); Fax: +133-4546-670 (H.L.)
†    These authors contributed equally to this work.

**Abstract:** Due to the changing climate, frequent episodes of drought have threatened cotton lint yield by offsetting their physiological and biochemical functioning. An efficient use of irrigation water can help to produce more crops per drop in cotton production systems. We assume that an optimal drip irrigation with low mepiquat chloride application could increase water productivity (WP) and maintain lint yields by enhancing leaf functional characteristics. A 2-year field experiment determines the response of irrigation regimes (600 ($W_1$), 540 ($W_2$), 480 ($W_3$), 420 ($W_4$) 360 ($W_5$) m$^3$ ha$^{-1}$) on cotton growth, photosynthesis, fiber quality, biomass accumulation and yield. Mepiquat chloride was sprayed in different concentration at various growth phases (see material section). Result showed that $W_1$ increased leaf area index (LAI) by 5.3–36.0%, net photosynthetic rate (Pn) by 3.4–23.2%, chlorophyll content (Chl) by 1.3–12.0% than other treatments. Improvements in these attributes led to higher lint yield. However, no differences were observed between $W_1$ and $W_2$ in terms of lint and seed cotton yield, but $W_2$ increased WP by 3.7% in both years. Compared with other counterparts, $W_2$ had the largest LAI (4.3–32.1%) at the full boll stage and prolonged reproductive organ biomass (ROB) accumulation by 30–35 d during the fast accumulation period (FAP). LAI, the average ($V_T$) and maximum ($V_M$) biomass accumulation rates of ROB were positively correlated with lint yield. In conclusion, the drip irrigation level of 540–600 m$^3$ ha$^{-1}$ with reduced MC application is a good strategy to achieve higher WP and lint yield by improving leaf photosynthetic traits and more reproductive organ biomass accumulation.

**Keywords:** drip irrigation quota; cotton; lint yield; water productivity; biomass

## 1. Introduction

Cotton (*Gossypium hirsutum* L.) is an important fiber crop and oil seed crop worldwide [1]. China produces an average lint yield of 1200 kg ha$^{-1}$, which is higher than India, Pakistan and USA [2]. With the increasing population comes an increased demand for fiber, and changes in climatic conditions are threating cotton productivity [3]. Crop intensification to produce more food, fiber and feed requires more water, but water resources are limited. Although, cotton is considered drought-resistant crop and

its productivity is negatively affected by drought stress. This can lead to reduced growth by negatively influencing plant physiological, biochemical and molecular events [2]. Drought stress can cause 50% to 73% reductions in cotton yields [4]. Limited water availability has threatened irrigated cotton production. Oh the other hand, sufficient fertilizer and irrigation supply results in luxury vegetative growth and increase insect pest incidence which lead to yield penalty [5]. In this context, there is a need to develop water conservation strategy to achieve more crops per drop [6].

Photosynthesis is the prerequisite for lint yield formation. The crops photosynthetic ability can be improved by regulating plant function and irrigation conditions, which in turn affect lint yield [7,8]. Deficit water affects biomass distribution and facilitates assimilate transfer to reproductive organs [9]. A short period of mild drought may stimulate the compensatory effect of photosynthesis [10]. These compensations favor the translocation of assimilate to reproductive organs and the improvement of WP (water productivity) without sacrificing yield [11]. Hence, these compensatory effects represent a self-regulatory mechanism that helps crop to adapt stressful environment by efficient utilization of limited water resources [1].

Xinjiang is the major cotton-growing provinces in China, contributing 67% of the total national lint production [12]. However, low water availability has imposed a great challenge to cotton production in this area. Currently, mulch drip irrigation is widely adopted to increase cotton lint yield and WP in Xinjiang [13,14]. To conserve water and produce high yields under irrigation systems, cotton growers have adopted the concept of regulated deficit irrigation (RDI) [10,13,15]. RDI can increase WP rapidly, this reduces leaf area and leading to lower photosynthesis rate [15,16], it was not conducive to biomass accumulation, resulting in failure to increase production [17,18]. Therefore, understanding the changes in photosynthetic characteristics and dry matter accumulation are needed to achieve optimal lint yield and WP under mulch drip irrigation system.

The main purpose of mulch drip irrigation technology is to conserve soil water and achieve high crop yield, but yield and productivity do not always increase with increased irrigation quota [19,20]. Hence, a reasonable control of drip irrigation quotas is essential to identify the optimal combination of water conservation and high yield. Mepiquat chloride (MC) (N,N-dimethylpiperidinium chloride) is a growth regulator used worldwide to control plant geometry. MC can be absorbed by leaves and is distributed throughout plants [21]. MC applications induce reductions in leaf expansion, stems, petiole length, and node number and enhance the maturity of cotton crops, with variable yield responses [22–24]. Therefore, it is hypothesized that moderately reduced irrigation quotas in conjunction with a low application rate of MC in the field can improve WP and maintain lint yields by utilizing the compensatory effects of photosynthesis under deficient drip irrigation. The objectives of this research were to explore the effects of various drip irrigation quotas on the photosynthesis capability, biomass accumulation, yield and WP using different concentration of MC under mulch drip irrigation systems. It also determines the quantitative relationships among these factors.

## 2. Materials and Methods

### 2.1. Experimental Site and Cultivar

Field experiments were conducted in 2016 and 2017 at the experimental farm of Shihezi University (45°19′ N latitude, 86°03′ E longitude). A Xinluzao 45 (*Gossypium hirsutum* L.) cultivar was used in this study. This cultivar was developed by the Xinjiang Academy of Agricultural and Reclamation Science and is officially registered and released by the Xinjiang Crop Cultivar Registration Committee. The total growth period from emergence to initial boll opening (BO) is 122 days. The soil was a purple clay loam with a pH of 7.65 and contained 15.3 g kg$^{-1}$ organic matter, 1.1 g kg$^{-1}$ total N, 54.9 mg kg$^{-1}$ available N, 23.0 mg kg$^{-1}$ available P and 194 mg kg$^{-1}$ available K within the 0–20 cm soil layer. The evapotranspiration, temperature and precipitation data from April to October are shown in Figure 1.

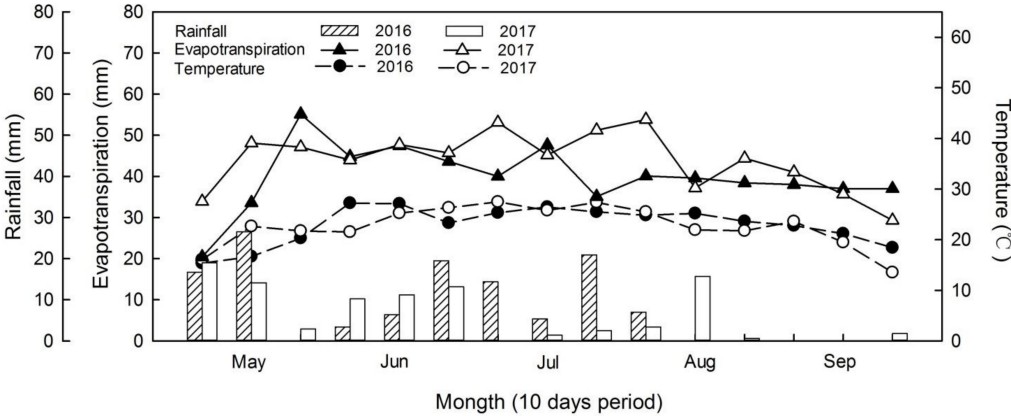

**Figure 1.** Monthly average evapotranspiration, temperature and rainfall of Shihezi (2016–2017).

### 2.2. Experimental Design

A randomized complete block design with four replications was used in this study. Generally, a total irrigation amount of 4800–5000 $m^3$ $ha^{-1}$ is required to achieve more than 6000 kg $ha^{-1}$ of seed cotton yield in northern Xinjiang region [10,18]. Five drip irrigation treatments were targeted i.e., $W_1$ (600 $m^3$ $ha^{-1}$ of water each time with the total amount of irrigation 4800 $m^3$ $ha^{-1}$; control), $W_2$ (540 $m^3$ $ha^{-1}$ of water each time with the total amount of irrigation 4320 $m^3$ $ha^{-1}$), $W_3$ (480 $m^3$ $ha^{-1}$ of water each time with the total amount of irrigation 3840 $m^3$ $ha^{-1}$), $W_4$ (420 $m^3$ $ha^{-1}$ of water each time, with the total amount of irrigation 3360 $m^3$ $ha^{-1}$), and $W_5$ (360 $m^3$ $ha^{-1}$ of water each time with the total amount of irrigation 2880 $m^3$ $ha^{-1}$). The drip irrigation rates were controlled by water meter and switch ball valve. The irrigation was applied in the same dates for all the treatments, and the duration was approximately 10–14 h (07:30 AM–21:30 PM).

### 2.3. Field Management

Prior to sowing, the experimental field was covered with a plastic film. Two drip irrigation lines (Beijing Luckrain Inc., China) were installed under each plastic film. The drip irrigation line had an inner diameter of 2.5 cm with emitter distance of 50 cm, and a flow rate of 2.7 L $h^{-1}$. Cotton seeds were sown on both sides of the drip irrigation belt at a distance of 13.5 cm on 21 and 23 April in 2016 and 2017, respectively. The plots were randomly arranged with the total area of 56 $m^2$ (7.0 × 8.0 $m^2$). After two weeks, seedlings were thinned to maintain the desired planting density. The row spacing was maintained as 12 cm with a planting density of 18,000 plants $ha^{-1}$ which was commonly practiced in this region. Fertilizer was applied with water by 8 times (first via drip irrigation for half an hour, then via fertigation). Thereafter, the field was fertilized with 4500 kg $ha^{-1}$ of oil residue (with 13% N, 2% $P_2O_5$ and 16% $K_2O$) as a basal fertilizer. In addition, 72 kg $ha^{-1}$ of urea (comprising 46% N) and 225 kg $ha^{-1}$ of triple superphosphate (comprising 45% $P_2O_5$) were applied throughout the growth period. The amount and time of the drip irrigation was controlled to maintain to distribute equal amount of fertilizer for each treatment. MC was applied to control vegetative growth. MC solution at 208 g $hm^{-2}$ concentration was sprayed 5 times in the $W_1$ treatment. A 6 g $hm^{-2}$ MC solution was sprayed from cotyledon stage to the two-leaf stage and 11 g $hm^{-2}$ was sprayed at the 5–7-leaf stage. Moreover, 26 g $hm^{-2}$, 45 g $hm^{-2}$, 120 g $hm^{-2}$ was sprayed 2 days before the first irrigation, 2 days before the second irrigation and 5–7 days after topping, respectively. The first and second spray was similar to $W_1$ treatment. However, an MC solution at a concentration of 137 g $hm^{-2}$ was sprayed on $W_2$, $W_3$, $W_4$ and $W_5$ treatments. To hasten the crop maturity, a defoliant at 450 g $ha^{-1}$ tribenuron combined with 1350 mL $ha^{-1}$ ethephon was used in both years. Artificial topping was carried out on 3 and 8 July in 2016 and 2017, respectively. Other management practices such as insect and weed control were conducted according to the local agronomic practices.

### 2.4. Net Photosynthetic Rate

Net photosynthetic rate (Pn) was assessed from functional leaf at the full squaring (FS), initial flowering (IF), full flowering (FF), full boll setting (FB), late boll setting (LFB) and BO stages between 10:00 a.m. and 12:00 p.m. on sunny day using an open-type photosynthesis system (LI-6400, LI-COR, Lincoln, NE, USA) equipped with a red/blue light source chamber. The machine was configured at light intensity of 1800 µmol m$^{-2}$ s$^{-1}$, temperature, 32 °C and at 20% relative humidity. Four plants in each plot were selected for measurement.

### 2.5. Water Productivity (WP)

The WP was determined according to [6],

$$WP = Y/I \tag{1}$$

WP, Y and I represent the water productivity (kg m$^{-3}$), seed cotton yield (kg ha$^{-1}$) and the amount of drip irrigation (m$^{-3}$) during the whole cotton growth period, respectively.

### 2.6. Chlorophyll Content

To measure chlorophyll content leaves were removed and the petiole was wrapped in wet gauze. A 0.1 g leaf sample was used to determine the Chl content. Leaves were placed in a 25 mL test tube and the pigment was extracted with 13 mL of 80% acetone. Tubes were wrapped with a black cloth and placed in dark conditions. Tubes were shaken at regular intervals and incubated for 72 h until the leaves become white color. The optical density (OD) value was measured at 470 nm, 663 nm, and 645 nm wavelength using a UV-2041 spectrophotometer (Shimadzu, Kyoto, Japan).

### 2.7. Biomass Accumulation

Four successive plants at the FS, IF, FF, FB, LFB and BO stages from each plot of the fourth replicates were carefully uprooted, divided into vegetative organs (roots, stem, leaves and branches) and reproductive organs (buds, flowers, boll shells and bolls). Samples were enveloped separately and placed into an electric fan-amended oven at 105 °C for 30 min then dried at 80 °C to a constant weight. The leaf area was measured using an LI-3000 area meter (LI-COR, Lincoln, NE, USA). Leaf area index (LAI) was calculated by multiplying the total leaf area of single plant (m$^2$ plant$^{-1}$) × plant density (plants m$^{-2}$). The declining rate of LAI, Pn and Chl content at the FB and BO stages was determined as follows:

$$\text{Declining Rate (\%)} = -(V_{BO} - V_{FB})/V_{BO} \tag{2}$$

Of which, $V_{BO}$ and $V_{FB}$ represent the LAI, Pn and Chl content parameters at the FB and BO stages, respectively.

A logistic formula was used to describe the progress of biomass accumulation [25]:

$$Y = \frac{K}{1 + ae^{bt}} \tag{3}$$

Of which, $t$ (day) indicates days after emergence (DAE), $Y$ (g) indicates the biomass at $t$, $K$ (g) is the maximum biomass, and $a$ and $b$ are the constants to be found.

$$t_0 = \frac{\ln a}{b} \tag{4}$$

$$t_1 = \frac{1}{b} \ln\left(\frac{2 + \sqrt{3}}{a}\right) \tag{5}$$

$$t_2 = \frac{1}{b}\ln\left(\frac{2 - \sqrt{3}}{a}\right) \tag{6}$$

At $t = t_0$, the biomass accumulation reaches a maximum speed defined as follows:

$$V_M = \frac{-bk}{4} \tag{7}$$

The period at which 58% of the biomass accumulated is defined as the biomass fast accumulation period (FAP), which begins at $t_1$ and terminates at $t_2$. During the FAP, Y is linearly correlated with $t$ and the average speed, defined as follows:

$$V_T = \frac{Y_2 - Y_1}{t_2 - t_1} \tag{8}$$

*2.8. Yield, Yield Contributors and Fiber Quality*

Seed cotton from each plot was hand-picked (on 3 and 15 October in 2016; 30 September and 15 October in 2017). Seed cotton was sun dried and weighed. One hundred fully opened bolls were sampled to calculate individual boll weight and lint percentage lint percentage. Boll number were determined by counting bolls (>2 cm in diameter) of each plant on 15 September and 20 September in 2016 and 2017, respectively.

To assess fiber quality attributes (length, strength, micronaire, and uniformity) lint samples were sent to the Chinese Academy of Agricultural Sciences for high-volume instrumentation analysis.

*2.9. Data Analysis*

SPSS 19.0 software (SPSS Institute Inc., Chicago, IL, USA) was used for analysis of variance. Means were tested by Duncan multiple comparison at a level of 0.05. Sigma Plot 12.5 (Aspire Software Intl., Ashburn, VA, USA) was used for data processing and figures as well as linear regression.

## 3. Results

*3.1. Leaf Area Index*

The LAI decreased with the decreasing drip irrigation level (except the FB stage) in both years (Figure 2). Under W1, the LAI was 5.6 at the LFB stage, 3.6 to 5.3 at the FB stage under for $W_5$, $W_4$, $W_3$ and $W_2$ treatments. At the FB stage, the LAI was 3.1–5.9% higher in $W_2$ compared with $W_1$. Moreover, $W_1$, $W_2$, $W_3$, $W_4$ and $W_5$ decreased the LAI by 5.7%, 14.6%, 18.6%, 18.6% and 18.7%, respectively, at all growth stages.

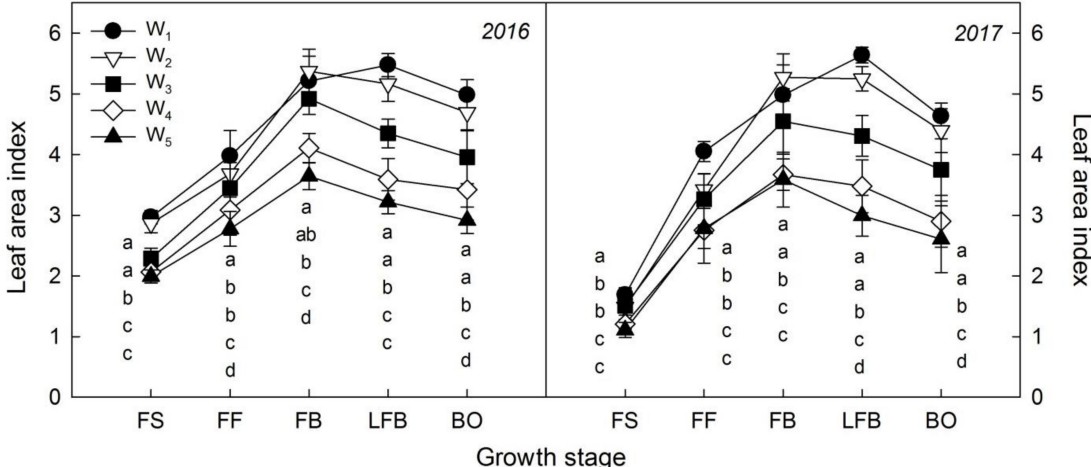

**Figure 2.** Effect of different drip irrigation quotas on leaf area index (LAI) of cotton at full squaring (FS), full flowering (FF), full boll (FB), later full boll (LFB) and boll opening stage (BO) in 2016 and 2017. Error bar shows standard error (SE) of means.

*3.2. Chlorophyll Content*

Cotton leaf Chl content was significantly influenced by irrigation levels at different growth stages (Figure 3). with the decreased in the irrigation level Chl content was significantly decreased at various growth stages. $W_1$ and $W_2$ had higher leaf Chl contents at FB stage and then decreased later in season. $W_2$–$W_5$ decreased the Chl by 0.4–5.2% at the FS stage, −0.11–9.7% at the FF stage, 7–16.6% at the FB stage, 1.0–6.3% at the LFB stage and 3.4–21.7% at the BO stage.

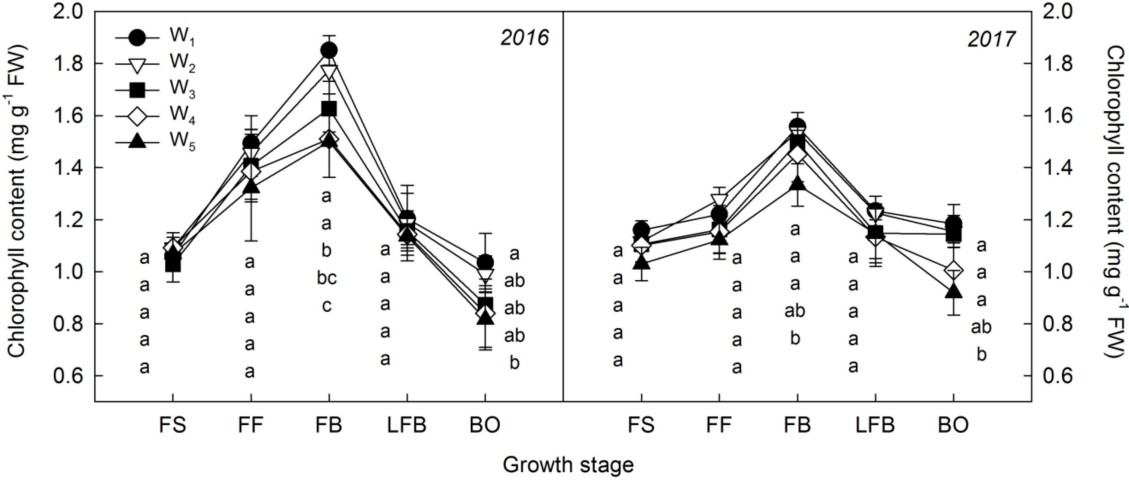

**Figure 3.** Effect of different drip irrigation quotas on chlorophyll (Chl) contents of cotton at full squaring (FS), full flowering (FF), full boll (FB), later full boll (LFB) and boll opening stage (BO) in 2016 and 2017. Error bar shows standard error (SE) of means.

### 3.3. Net Photosynthetic Rate

The Pn rate was substantially influenced by irrigation levels. With the crop development the Pn rate was increased and then decreased (Figure 4). The rate of Pn was lowered with the decreased of irrigation level during the whole growth stages. $W_1$ and $W_2$ resulted in higher net Pn compared with other counterparts. Across the years, the Pn was higher in 2016 compared with 2017.

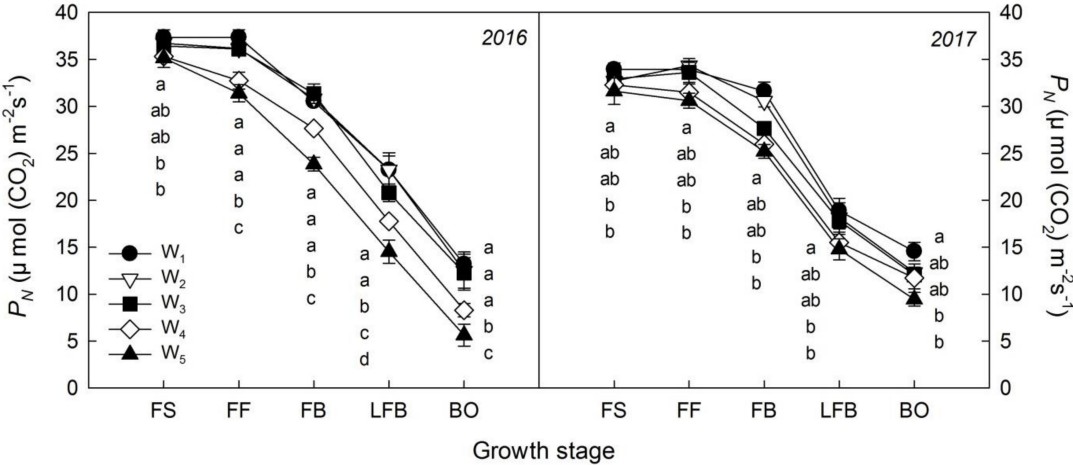

**Figure 4.** Effect of different drip irrigation quotas on net photosynthetic rate (Pn) of cotton leaf at full squaring (FS), full flowering (FF), full boll (FB), later full boll (LFB) and boll opening stage (BO) in 2016 and 2017. Error bar shows standard error (SE) of means.

### 3.4. Cotton Plant Biomass Accumulation

Cotton plant biomass (CPB) accumulation was increased rapidly and then decreased with decreasing drip irrigation (Figure 5). $W_1$ resulted in 7.84%, 13.13%, 22.17%, and 28.09% at the FB stage; 8.24%, 15.80%, 27.82%, and 33.91% at the LFB stage; and 9.72%, 15.60%, 27.54%, and 34.09% higher biomass at the BO stage averaged across both years. Vegetative organ biomass (VOB) accumulation increased sharply before the FF stage and then decreased with decreasing drip irrigation (Figure 5). $W_1$ increased the VOB by 6.97–32.97% at the FB stage and 5.86–34.20% at the LFB stage than other treatment. Reproductive organ biomass (ROB) accumulation decreased with decreasing drip irrigation (Figure 5). $W_1$ had higher ROB by 15.32% and 11.04% at the LFB stage and 17.21% and 18.74% at the BO stage, respectively.

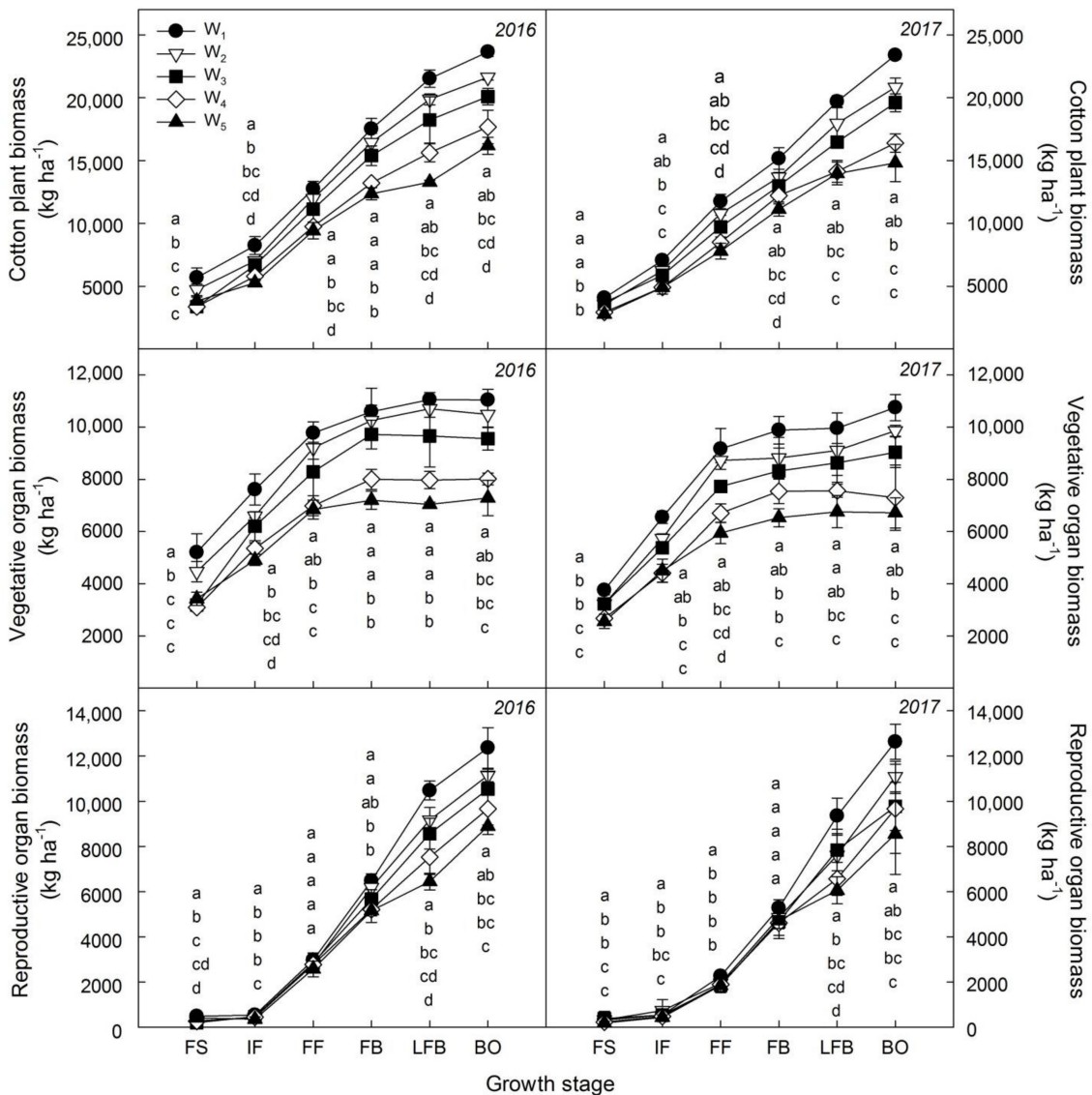

**Figure 5.** Response of cotton plant biomass (CPB), vegetative organs biomass (VOB), and reproductive organ biomass (ROB) at full squaring (FS), initial flowering (IF), full flowering (FF), full boll (FB), later full boll (LFB) and boll opening stage (BO) under drip irrigation quotas in 2016 and 2017. Error bar shows standard error (SE) of means.

### 3.5. Characteristics of Biomass Accumulation

The simulation of biomass as a function of DAE was assessed via equation (2). The logistic function of the biomass accumulation was followed by a sigmoidal growth pattern. All the coefficients of determination were significant, although the equation coefficients differed among the treatments (Table 1). Calculations by Equations (3)–(8) based on Table 1 revealed the beginning and end day of the FAP for CPB accumulation during both years. $W_1$ and $W_2$ begins and ends at 68 and 119 DAE and 69 and 113 DAE, respectively, in 2016 and 64 and 124 DAE and 77 and 115 DAE, respectively, in 2017, with greater average and maximum rates of biomass over other treatments.

**Table 1.** Eigen parameters of cotton biomass accumulations for different drip irrigation quotas.

| Year | Treatment | Regression Equations | $p$ | Fast Accumulation Period | | | | Fast Accumulation Point | |
|---|---|---|---|---|---|---|---|---|---|
| | | | | $t_1$ (DAE) | $t_2$ (DAE) | T (day) | $V_T$ (kg ha$^{-1}$ day$^{-1}$) | $V_M$ (kg ha$^{-1}$ day$^{-1}$) | $t_m$ (DAE) |
| | Cotton plant biomass | | | | | | | | |
| | $W_1$ | $Y = 33258.5934/(1 + 119.3122e^{-0.051359t})$ | 0.0011 | 67.5 | 118.8 | 51.3 | 374.4 | 427.0 | 93.1 |
| | $W_2$ | $Y = 29446.6185/(1 + 231.5378e^{-0.05993t})$ | 0.0016 | 68.9 | 112.8 | 44.0 | 386.8 | 441.2 | 90.9 |
| | $W_3$ | $Y = 25670.201/(1 + 403.4676e^{-0.067762t})$ | 0.0020 | 69.1 | 108.0 | 38.9 | 381.3 | 434.9 | 88.5 |
| | $W_4$ | $Y = 22840.8431/(1 + 299.7992e^{-0.063716t})$ | 0.0026 | 68.8 | 110.2 | 41.3 | 319.0 | 363.8 | 89.5 |
| | $W_5$ | $Y = 20179.7177/(1 + 237.2227e^{-0.062861t})$ | 0.0059 | 66.1 | 108.0 | 41.9 | 278.1 | 317.1 | 87.0 |
| | Vegetative organ biomass | | | | | | | | |
| 2016 | $W_1$ | $Y = 13417.5274/(1 + 2823.3601e^{-0.114252t})$ | 0.0007 | 58.0 | 81.1 | 23.1 | 336.0 | 383.2 | 69.5 |
| | $W_2$ | $Y = 13262.6927/(1 + 1995.0725e^{-0.105198t})$ | 0.0007 | 59.7 | 84.8 | 25.0 | 305.8 | 348.8 | 72.2 |
| | $W_3$ | $Y = 11790.2110/(1 + 14040.0792e^{-0.131561t})$ | 0.0019 | 62.6 | 82.6 | 20.0 | 340.0 | 387.8 | 72.6 |
| | $W_4$ | $Y = 9872.6580/(1 + 11420.6305e^{-0.129823t})$ | 0.0019 | 61.8 | 82.1 | 20.3 | 280.9 | 320.4 | 72.0 |
| | $W_5$ | $Y = 8836.4909/(1 + 14624.8702e^{-0.136997t})$ | 0.0007 | 60.4 | 79.6 | 19.2 | 265.4 | 302.6 | 70.0 |
| | Reproductive organ biomass | | | | | | | | |
| | $W_1$ | $Y = 17786.1982/(1 + 7812.2131e^{-0.083488t})$ | 0.0012 | 91.6 | 123.1 | 31.6 | 325.5 | 371.2 | 107.4 |
| | $W_2$ | $Y = 15248.9874/(1 + 9516.8349e^{-0.086831t})$ | 0.0024 | 90.3 | 120.7 | 30.3 | 290.2 | 331.0 | 105.5 |
| | $W_3$ | $Y = 14945.7964/(1 + 5473.7872e^{-0.080545t})$ | 0.0033 | 90.5 | 123.2 | 32.7 | 263.9 | 301.0 | 106.9 |
| | $W_4$ | $Y = 14135.8253/(1 + 3096.8490e^{-0.074301t})$ | 0.0053 | 90.5 | 125.9 | 35.5 | 230.2 | 262.6 | 108.2 |
| | $W_5$ | $Y = 12496.6137/(1 + 2551.9707e^{-0.073465t})$ | 0.0109 | 88.9 | 124.7 | 35.9 | 201.2 | 229.5 | 106.8 |

**Table 1.** *Cont.*

| Year | Treatment | Regression Equations | $p$ | Fast Accumulation Period | | | | Fast Accumulation Point | |
|---|---|---|---|---|---|---|---|---|---|
| | | | | $t_1$ (DAE) | $t_2$ (DAE) | T (day) | $V_T$ (kg ha$^{-1}$ day$^{-1}$) | $V_M$ (kg ha$^{-1}$ day$^{-1}$) | $t_m$ (DAE) |
| 2017 | Cotton plant biomass | | | | | | | | |
| | $W_1$ | $Y = 35207.8503/(1 + 60.9476e^{-0.043942t})$ | 0.0017 | 63.6 | 123.5 | 59.9 | 339.1 | 386.8 | 93.5 |
| | $W_2$ | $Y = 30621.6718/(1 + 67.5922e^{-0.046357t})$ | 0.0017 | 62.5 | 119.3 | 56.8 | 311.2 | 354.9 | 90.9 |
| | $W_3$ | $Y = 28700.9055/(1 + 60.3377e^{-0.044661t})$ | 0.0015 | 62.3 | 121.3 | 59.0 | 281.0 | 320.5 | 91.8 |
| | $W_4$ | $Y = 22741.8283/(1 + 106.6472e^{-0.054536t})$ | 0.0013 | 61.5 | 109.8 | 48.3 | 271.9 | 310.1 | 85.6 |
| | $W_5$ | $Y = 20470.5648/(1 + 88.5934e^{-0.053107t})$ | 0.0021 | 59.6 | 109.2 | 49.6 | 238.3 | 271.8 | 84.4 |
| | Vegetative organ biomass | | | | | | | | |
| | $W_1$ | $Y = 12345.7473/(1 + 701.2048e^{-0.108047t})$ | 0.0003 | 48.5 | 72.8 | 24.4 | 292.4 | 333.5 | 60.6 |
| | $W_2$ | $Y = 11733.5035/(1 + 613.4740e^{-0.102951t})$ | 0.0020 | 49.6 | 75.1 | 25.6 | 264.8 | 302.0 | 62.4 |
| | $W_3$ | $Y = 10606.1122/(1 + 356.4513e^{-0.096032t})$ | 0.0022 | 47.5 | 74.9 | 27.4 | 223.3 | 254.6 | 61.2 |
| | $W_4$ | $Y = 9074.6476/(1 + 718.0429e^{-0.106314t})$ | 0.0016 | 49.5 | 74.2 | 24.8 | 211.5 | 241.2 | 61.9 |
| | $W_5$ | $Y = 8327.4903/(1 + 370.5410e^{-0.097825t})$ | 0.0019 | 47.0 | 73.9 | 26.9 | 178.6 | 203.7 | 60.5 |
| | Reproductive organ biomass | | | | | | | | |
| | $W_1$ | $Y = 18820.0391/(1 + 3123.6047e^{-0.076271t})$ | 0.0004 | 88.2 | 122.8 | 34.5 | 314.6 | 358.9 | 105.5 |
| | $W_2$ | $Y = 16606.3387/(1 + 2573.79391e^{-0.074759t})$ | 0.0027 | 87.4 | 122.7 | 35.2 | 272.1 | 310.4 | 105.0 |
| | $W_3$ | $Y = 15124.7130/(1 + 2981.4888e^{-0.077040t})$ | 0.0010 | 86.7 | 120.9 | 34.2 | 255.4 | 291.3 | 103.8 |
| | $W_4$ | $Y = 13679.9428/(1 + 2689.8974e^{-0.077117t})$ | 0.0056 | 85.3 | 119.5 | 34.2 | 231.2 | 263.7 | 102.4 |
| | $W_5$ | $Y = 9488.9482/(1 + 7536.0818e^{-0.09532t})$ | 0.0050 | 79.8 | 107.5 | 27.6 | 198.3 | 226.1 | 93.7 |

DAE means days after emergence (day); $t_1$ and $t_2$ are the beginning and terminating day after the fast accumulation period; T indicates the duration of FAP, T = $t_1$ − $t_2$; $V_T$ and $V_M$ are the average and maximum biomass accumulation rates during the FAP, respectively.

Differences in CPB accumulation were noticed among the treatments in both years (Table 1). $W_1$ had the greatest average (356.8 kg $ha^{-1}$ $day^{-1}$ $V_T$) and maximum (406.9 kg $ha^{-1}$ $day^{-1}$ $V_M$) biomass accumulation rates followed by $W_2$ (349.0 kg $ha^{-1}$ $day^{-1}$ $V_T$, 398.1 kg $ha^{-1}$ $day^{-1}$ $V_M$) and $W_3$ (331.0 kg $ha^{-1}$ $day^{-1}$ $V_T$, 377.7 kg $ha^{-1}$ $day^{-1}$ $V_M$). The FAP of the CPB accumulation under $W_2$ and $W_3$ initiated at the same time (65.7 DAE), and they terminated 5 days and 7 days sooner than that under $W_1$. $W_5$ had a shorter duration of CPB at FAP which showed lowest average (258.2 kg $ha^{-1}$ $day^{-1}$ $V_T$) and maximum (294.5 kg $ha^{-1}$ $day^{-1}$ $V_M$) biomass accumulation rates.

The VOB accumulation was also affected by the irrigation quota during both years (Table 1). The $V_T$ and $V_M$ increased with increasing drip irrigation in both years. VOB accumulation at FAP under $W_2$ and $W_3$ begun and terminated almost at the same time, which was 1–3 days delayed than $W_1$. $W_2$ had longer VOB duration at FAP (25.3 days) with both average ($V_T$ 285.3 kg $ha^{-1}$ $day^{-1}$) and maximum ($V_M$ 325.4 kg $ha^{-1}$ $day^{-1}$) biomass formation rates.

The drip irrigation quotas significantly altered cotton plant ROB organ biomass formation in both years (Table 1). Under, the FAP of $W_2$ ROB accumulation at FAP began at 89 DAE and terminated at 122 DAE, both of which 1 day delayed than $W_1$. Moreover, $W_2$ and $W_3$ had similar FAP time. $W_1$ had higher both average and maximum biomass accumulation rates at FAP followed by $W_2$.

*3.6. Yield, Water Productivity and Fiber Quality*

Cotton yield and yield components were significantly influenced by irrigation levels (Table 2). Among irrigation levels $W_1$ produced the highest seed cotton and lint yield in both years compared with other treatments. Compared with $W_1$, $W_2$ slightly influenced the cotton yield in 2016 and 2017. $W_2$ resulted in 5.3–7.7% lower seed cotton yield and a 5.0–5.7% lower lint yield. Compared with $W_1$, individual boll weight was decreased by 1.6%, 2.4%, 4.1%, and 5.3% $W_2$, $W_3$, $W_4$, and $W_5$ respectively. Similarly boll numbers per unit area decreased by 5.4%, 6.2%, 9.9%, and 18.4%, respectively. However the lint % increased. In addition, WP under $W_2$, $W_3$, $W_4$, and $W_5$ was 3.9%, 13.0%, 23.6%, and 29.8% greater than over $W_1$. The differences were minor between both years.

**Table 2.** Cotton yield and WP under different drip irrigation quotas.

| Year | Treatment | Seed Yield (kg $ha^{-1}$) | Lint Yield (kg $ha^{-1}$) | Boll Weight (g) | Bolls Per Unit Area ($10^4$ $ha^{-1}$) | Lint Percentage (%) | Water Productivity (kg $m^{-3}$) |
|---|---|---|---|---|---|---|---|
| 2016 | $W_1$ | 6607 ± 392 a | 2739 ± 194 a | 4.78 ± 0.12 a | 140.2 ± 11.4 a | 41.45 ± 1.43 b | 1.38 ± 0.08 b |
| | $W_2$ | 6099 ± 305 ab | 2581 ± 109 ab | 4.72 ± 0.07 ab | 129.2 ± 5.9 ab | 42.32 ± 1.83 ab | 1.41 ± 0.07 b |
| | $W_3$ | 5968 ± 286 ab | 2531 ± 212 ab | 4.69 ± 0.14 ab | 129.8 ± 9.0 ab | 42.41 ± 1.84 ab | 1.55 ± 0.07 ab |
| | $W_4$ | 5694 ± 340 b | 2393 ± 235 bc | 4.62 ± 0.19 bc | 123.1 ± 9.5 bc | 42.39 ± 1.82 ab | 1.68 ± 0.10 a |
| | $W_5$ | 5013 ± 260 c | 2147 ± 85 c | 4.55 ± 0.17 c | 110.2 ± 4.7 c | 42.84 ± 1.85 a | 1.74 ± 0.09 a |
| 2017 | $W_1$ | 6492 ± 466 a | 2615 ± 199 a | 4.76 ± 0.16 a | 135.9 ± 12.1 a | 40.28 ± 1.39 b | 1.35 ± 0.10 c |
| | $W_2$ | 6151 ± 341 ab | 2484 ± 96 ab | 4.67 ± 0.26 ab | 132.0 ± 10.7 ab | 40.39 ± 1.75 b | 1.42 ± 0.08 bc |
| | $W_3$ | 5874 ± 441 bc | 2397 ± 248 b | 4.62 ± 0.15 ab | 129.3 ± 11.1 ab | 40.63 ± 1.76 ab | 1.53 ± 0.11 b |
| | $W_4$ | 5689 ± 342 c | 2358 ± 233 b | 4.53 ± 0.09 b | 125.6 ± 7.5 b | 41.45 ± 1.79 a | 1.69 ± 0.10 a |
| | $W_5$ | 5184 ± 533 d | 2132 ± 165 c | 4.49 ± 0.11 b | 115.2 ± 14.4 c | 41.13 ± 1.78 ab | 1.80 ± 0.18 a |
| | Year | ns | * | ns | ns | ** | ns |
| | Year × Treatment | ns | ns | ns | ns | ns | ns |

Means within a column of the same year followed by a different letter are significantly different ($p < 0.05$) according to the Duncan multiple range test. The same letters in the same column indicated no significant difference at 0.05 level in Duncan's analysis in the same year. "*", "**" means significance at the 0.05, 0.01 level, respectively. "ns" indicates non-significant.

Fiber quality parameters were substantially influenced by irrigation levels (Table 3). The fiber length and uniformity increased as the drip irrigation quota increased. Compared with $W_1$, $W_2$ and $W_3$ had higher fiber lengths and fiber uniformity compared with other treatment. $W_4$ and $W_5$ treatment resulted in significantly lower fiber length and uniformity during both years. The uniformity was significantly greater in 2016 than in 2017, while the fiber length, specific strength, and micronaire values remained similar.

**Table 3.** Change of the fiber quality attributes under different drip irrigation quotas.

| Year | Treatment | Fiber Length (mm) | Fiber Uniformity (%) | Specific Strength (CN tex$^{-1}$) | Micronaire Value |
|---|---|---|---|---|---|
| 2016 | $W_1$ | 30.4 ± 0.38 a | 87.3 ± 0.35 a | 30.5 ± 1.5 a | 4.0 ± 0.11 a |
| | $W_2$ | 30.0 ± 0.06 ab | 87.0 ± 0.69 a | 30.7 ± 1.5 a | 4.2 ± 0.15 a |
| | $W_3$ | 29.9 ± 0.38 ab | 86.8 ± 0.33 a | 31.2 ± 1.2 a | 4.1 ± 0.10 a |
| | $W_4$ | 29.6 ± 0.33 b | 85.9 ± 0.31 b | 30.8 ± 0.7 a | 4.2 ± 0.17 a |
| | $W_5$ | 29.5 ± 0.35 b | 85.8 ± 0.15 b | 30.7 ± 1.3 a | 4.2 ± 0.15 a |
| 2017 | $W_1$ | 30.6 ± 0.48 a | 85.1 ± 0.78 a | 30.7 ± 0.20 a | 4.0 ± 0.31 a |
| | $W_2$ | 30.2 ± 0.53 ab | 84.7 ± 0.50 ab | 30.8 ± 0.10 a | 4.0 ± 0.14 a |
| | $W_3$ | 30.2 ± 0.10 ab | 84.6 ± 0.62 ab | 30.7 ± 0.72 a | 4.1 ± 0.07 a |
| | $W_4$ | 29.5 ± 0.21 bc | 84.4 ± 0.19 ab | 30.7 ± 0.40 a | 4.0 ± 0.31 a |
| | $W_5$ | 29.0 ± 0.84 c | 84.2 ± 0.47 b | 30.6 ± 0.62 a | 4.1 ± 0.26 a |
| | Year | ns | ** | ns | ns |
| | Year × Treatment | * | ns | ns | ns |

Means within a column of the same year followed by a different letter are significantly different ($p < 0.05$) according to the Duncan multiple range test. The same letters in the same column indicated no significant difference at 0.05 level in Duncan's analysis in the same year. "*", "**" means significance at the 0.05, 0.01 level, respectively. "ns" indicates non-significant.

*3.7. Correlation Analysis and Regression Analysis*

The relationships between photosynthetic characteristic parameters (the LAI, Pn and Chl), biomass accumulation (CPB, VOB and ROB) and lint yield were analyzed at different growth stages in both years (Table 4). The correlation intensity between the photosynthetic characteristic and lint yield was determined i.e., Pn (0.915) > LAI (0.896) > Chl (0.840) at the FB stage as well as LAI (0.916) > Pn (0.901) > Chl (0.727) at the LFB stage. During the FB to BO stages, the CPB, VOB and ROB were highly significantly ($p < 0.001$) correlated with lint yield at especially at the LFB stage. In addition, the correlation association between LAI and ROB and lint yield was gradually increased from FS to BO stage.

**Table 4.** Correlation between physiological parameters and lint yield and at different growth stages.

| Growth Stages | LAI | Pn | Chl | CPB | VOB | ROB |
|---|---|---|---|---|---|---|
| FS | 0.637* | 0.577 | 0.324 | 0.722* | 0.748* | 0.291 |
| FF | 0.918** | 0.907** | 0.592 | 0.919** | 0.940** | 0.47 |
| FB | 0.896** | 0.915** | 0.840** | 0.924** | 0.964** | 0.704* |
| LFB | 0.916** | 0.901** | 0.727* | 0.959** | 0.964** | 0.944** |
| BO | 0.946** | 0.806** | 0.498 | 0.946** | 0.946** | 0.937** |

"*" and "**" means significance at the 0.05, 0.01 level, respectively (both sides).

The relationships from regression analysis showed a declining rate of photosynthetic characteristics traits (LAI, Pn and Chl) from FB to BO stages. The lint yield has been shown in Figure 6. Cotton biomass accumulation (CPB, VOB and ROB), simulation (T, $V_T$, $V_M$ and $t_m$) and lint yield (Figure 7) were described using linear functions during different growth stages in both years. The declined rate of Pn and Chl content during the FB to BO stages, the T of CPB, VOB and ROB accumulation and $t_m$ of VOB and ROB accumulation were not significantly linearly correlated to lint yield. A negative correlation was observed between the declining rate of LAI ($R^2 = 0.7429$, $p < 0.001$) and lint yield. A positive correlation were noticed between $V_T$ ($R^2 = 0.7422$, $p < 0.001$) and $V_M$ ($R^2 = 0.7424$, $p < 0.001$) of CPB, VOB and ROB i.e., $V_T$ ($R^2 = 0.5791$, $p = 0.0106$), $V_M$ ($R^2 = 0.5791$, $p = 0.0106$), $V_T$ ($R^2 = 0.9354$, $p < 0.001$), $V_M$ ($R^2 = 0.9354$, $p < 0.001$) and lint yield. In addition, CPB $t_m$ was also correlated ($R^2 = 0.6702$, $p = 0.0038$) with lint yield.

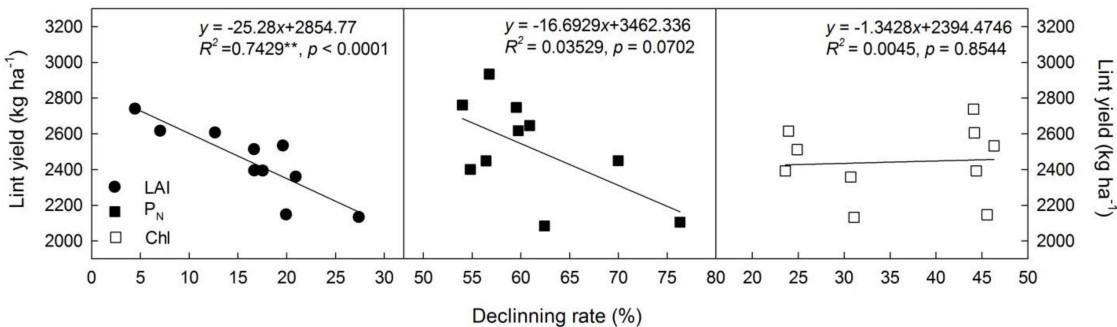

**Figure 6.** Regression analysis between the declining rate of photosynthetic capacity values (LAI, Pn and Chl) during FB to BO stage and lint yield. $R^2$ represents the coefficient of determination in linear regression. "**" means significance at the 0.01 level (both sides).

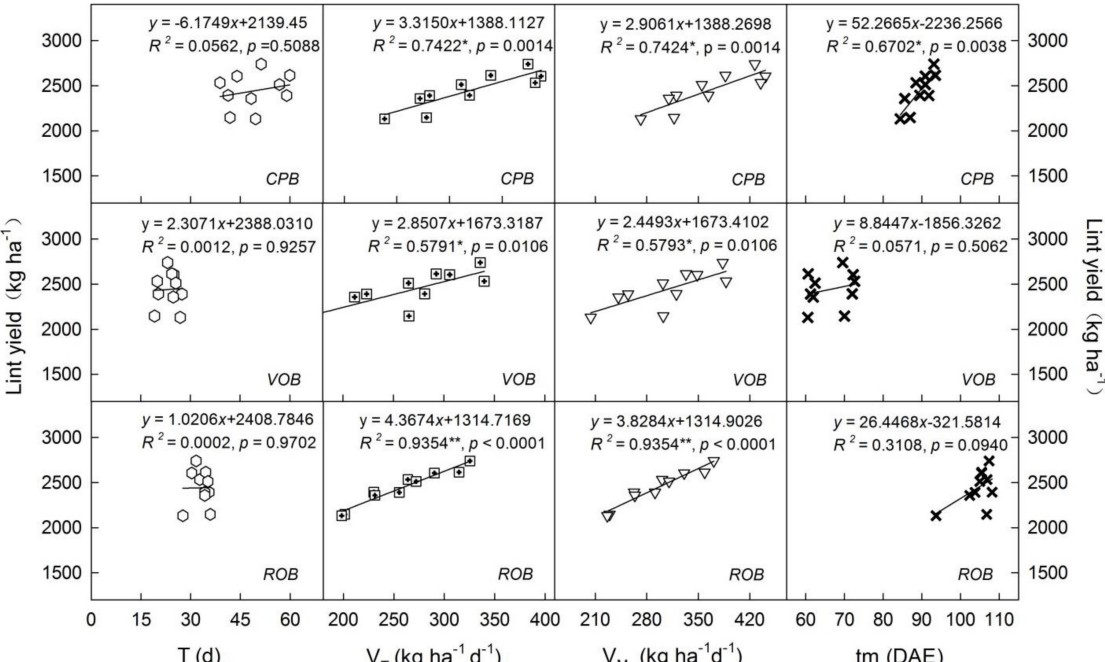

**Figure 7.** Regression analysis of the characteristic values (T, VT, VM and tm) of cotton biomass accumulation (CPB, VOB and ROB) and lint yield. T indicates the duration of FAP; $V_T$ and $V_M$ are the average and maximum biomass accumulation rates during the FAP, $t_m$ (DAE) means days after emergence (day) respectively. $R^2$ represents the coefficient of determination in linear regression. "*" and "**" means significance at the 0.05, 0.01 level, respectively (both sides).

## 4. Discussion

Crop production is positively allied with photosynthesis capacity (i.e., photosynthetic area, Pn, and photosynthetic pigments) [26,27] and is significantly influenced by soil water content [28]. In this study, LAI, Pn and Chl content were positively correlated with lint yield from full boll to boll opening stages. The LAI was strongly correlated with lint yield during the later full boll to boll opening stages and the declining rate was negatively correlated with lint yield. This show that duration of LAI during late growth late stages and the leaf photosynthetic capacity important players for increasing cotton yield. These data are in line with [29] that the absorption of photosynthetically active radiation was not significantly affected by mild water deficit. Plants can respond to drought by reducing nonstomatal transpiration (soil evaporation) [30] and increasing stomatal resistance (reducing evaporation) and osmotic adjustment substances [31]. An optimistic growth i.e., Pn, root growth, the LAI, plant height and biomass accumulation maintain high values in under short term water deficit which in turn

increase yield [10,32]. These physiological adjustments can be explained by the compensatory growth of cotton under moderate drought stress [33]. Hence, irrigation strategies can be used to alter leaf area expansion, the absorption of photosynthetically active radiation and carbohydrate production to enhance photosynthesis capacity, water conservation and consequently yield [27,29].

Photosynthesis is the basis of crop biomass accumulation and yield formation under drought conditions [34,35]. Chl affects electron transport and determines the photosynthesis capacity of crop plants as well plays a key role in the absorption, transmission and transformation of light energy [36]. In the present study, reductions in the Pn under water deficit conditions occurred due to Chl degradation [37]. This degradation may associate with low drip irrigation quota, increased stomatal resistance and low $CO_2$ supply to the chloroplast [38].

Leaf area is more sensitive to moisture stress compared with Pn and Chl [29]. A moderate reduction in drip irrigation quota is beneficial for low Chl and can delay leaf senescence. In this study, $W_2$ and $W_3$ had a negative effect on photosynthetic apparatus in the chloroplasts. This might be due to the change in the photosynthetic pigments or protection of the photosynthetic apparatus from photoinhibitory damage in the leaves [39]. However, the Calvin Cycle enzyme (ribulose-1,5-bisphosphate carboxylate/oxygenase, Rubisco) activity was maintained due to higher Pn and Chl content for biomass accumulation. The difference in dry matter was the result of size and duration of the photosynthetic area. $W_1$ delayed LAI which may be related to the exuberant development of leaves and the longer CPB accumulation time. $W_2$ maintained a relatively high LAI (>5.0) and a sufficient photosynthetic area resulted in assimilate formation and water conservation [30,40]. This phenomenon might be due to the growth compensatory effects of plants under slightly reduced irrigation quotas [41,42]. Plants can adapt to mild drought through various physiological activities such as increasing leaf area to maintain a favorable water content [43]. An optimal LAI of cotton plants lead to the absorption of sufficient light energy. This absorption improves both the population structure and canopy photosynthesis, thereby improving the light energy utilization and consequently high yield [44,45]. Furthermore, under a low level of MC application under $W_2$ treatment before the first and second irrigation events maintained a reasonable LAI and could create a reasonable population structure to guarantee a greater and more efficient photosynthetic system. An expansion of cotton leaves are considered more sensitive to drought than Pn [46]; this sensitivity could explain the significant decrease in photosynthetic area under $W_3$, $W_4$ and $W_5$ during the late growth period. Although, a lower LAI is conducive to light absorption within a lower canopy, it also decrease light energy and reduces yield [47]. This might be due to the lower irrigation quotas which did not provide suitable leaf moisture conditions. This further reduced LAI, increased the degradation rate of Chl and increased leaf senescence [48]. These alterations may affect integrity of the photosynthesis and reduced the photosynthetic efficiency [49].

Biomass accumulation is the final product of plant photosynthesis and more distribution of biomass to the reproductive organs are essential for high cotton yield [50]. More biomass accumulations are important to maintain high crop yields [51]. A significant or extremely significant positive correlation between biomass accumulation (VOB, VOB and ROB) and lint yield in the present study. Based on regression analysis, CPB, VOB and ROB biomass accumulation in both $V_T$ and $V_M$ were positively correlated to lint yield. Conversely, reductions in crop yield caused by irrigation have been attributed to decreased biomass formation [52]. Biomass accumulation at FAP was associated with increased water uptake. An appropriate irrigation quota could increase both average and maximum rates of CPB and can lead to increased biomass accumulation and consequently high yield [3]. In this study, $W_2$ and $W_3$ shortened CPB accumulation duration and facilitated maximum rates of CPB at FAP. However, $W_2$ had a longer duration of VOB accumulation. These conditions increased the distribution of assimilate to the reproductive organs. A moderate reduction in irrigation can maintain a high LAI to ensure high biomass accumulation [53]. This further transitioned more vegetative growth to reproductive growth and reduced evaporation during vegetative development [54]. Conversely, $W_4$ and $W_5$ significantly shortened the duration for biomass formation, thus reduced the maximum rates for VOB and ROB accumulation. This finding indicated that relatively low soil water contents are not good for growth

and development of the aboveground parts during the vegetative growth stage. These adverse effects may involve physiological responses [55], leaf area expansion [48], root growth [56] resulting in a decreased plant VOB to reproductive biomass and ultimately reduced yield [11]. Luxury, vegetative growth can consume excessive amount of nutrients and increases competition between vegetative and reproductive growth and consequently fruit shedding [57]. Together, these data showed that reducing irrigation quotas are not conducive to cotton growth or yield formation.

An appropriate irrigation level is important for sustainable cotton production in arid regions. Different irrigation amount can lead to significant differences in crop growth and both the accumulation and redistribution of photosynthesis assimilate [49], which in turn affects crop yield, water use efficiency and fiber quality [9,58]. A 15% reduction in the total irrigation amount can save irrigation water and reduced yield losses. However, further reduction up to 25% conserve more irrigation water but can lead high yield penalty [20]. Interestingly, the yield under $W_2$ was not significantly different from $W_1$ and the WP also increased in the present study. These results are consistent with those of previous research [30,59] who also reported that a slight reduction in drip irrigation can cause physiologically relevant adaptations in cotton, such as improved photosynthesis capabilities (leaf area and Chl content per unit area) [29] and growth promotion of vegetative organs [60]. Another possible reason might be due to reduced application of MC under $W_2$. The reduced use of MC in this treatment may facilitated vegetative growth [54] and increased the balance between vegetative and reproductive growth [61]. These phenomena were also beneficial for cotton plants in terms of maintaining a self-adjustment ability via the relationship between boll number and single boll weight [62]. Although, $W_4$ and $W_5$ presented relatively high WP but did not increase yield. $W_4$ and $W_5$ significantly reduced boll weight and bolls per unit area. This reduction in boll number under reduced irrigation further decreased lint yield. However, $W_2$ slightly reduced individual boll weight and number of bolls per unit area. This increment in yield maight be associated with a moderate reduction in the drip irrigation quota [9].

Cotton fiber length, fiber uniformity, specific strength and micronaire value are the important fiber quality parameters. In this study, moderately reduced drip irrigation ($W_2$ and $W_3$) quotas did not significantly affect cotton fiber quality parameters. These data are consistent with the results of previous studies [59,63]. However, the extremely low drip irrigation quota ($W_5$) significantly reduced fiber length and uniformity. The difference in fiber length may be due to moisture effects on fiber length which influence fiber elongation phase. The micronaire value is a measure of fiber fineness and maturity [64]. No significant differences in micronaire value or specific strength among the different treatments were observed. This might be related to the time interval of irrigation, which influenced cotton boll development.

## 5. Conclusions

In this study, irrigation quota and MC application had a significant effect on leaf photosynthetic performance and biomass accumulation, cotton yield, fiber quality and water productivity. Compared with $W_1$, $W_2$ had higher Pn and Chl content during all growth period. Moreover, $W_2$ combined with reduced MC application resulted in greater LAI at the full boll stage, which ensured a sufficient photosynthetic area and prolonged ROB accumulation duration and yield formation. In conclusion, the drip irrigation level of 540–600 m$^3$ ha$^{-1}$ with reduced MC application is a good strategy to maintain higher WP and achieve high lint yield as well as better fiber quality.

**Author Contributions:** H.L. and F.W. conceived and designed the experiments; J.X. and X.H. performed the experiments; H.G. and H.M. analyzed the data; A.K. edited and revised the paper; H.G. wrote the paper.

**Funding:** This project was supported by the National Key R&D Program of China (2017YFD0201900), National Natural Science Foundation of China (31760355), and Program of Youth Science and Technology Innovation Leader of The Xinjiang Production and Construction Corps (2017CB005).

**Conflicts of Interest:** The authors declare no conflict of interest. The funders had no role in the design of the study; in data collection, analyses, or interpretation of data; in the writing of the manuscript, or in the decision to publish the results.

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
