# Peer review of "Moderate Drip Irrigation Level with Low Mepiquat Chloride Application Increases Cotton Lint Yield by Improving Leaf Photosynthetic Rate and Reproductive Organ Biomass Accumulation in Arid Region"

_agronomy, doi:10.3390/agronomy9120834_

Round 1
Reviewer 1 Report
I think this is a good paper. However, the only problem I have with this paper is that the experiment consisted in applying different irrigation rates to evaluate the response to MC. However, the authors don't mention details about how irrigation was calculated. How they decided when to apply irrigation? How the different irrigation regimes were decided? It seems that the highest irrigation treatment was to apply 6 cm per irrigation with 8 irrigations. The lowest irrigation treatment was to apply an irrigation depth of 3.6 cm in 8 irrigation events. The irrigation was applied in the same dates for all the treatments? What was the evapotranspiration, and how much of that ET was supplied by irrigation?. What is the average annual rainfall for that area? Is it variable with time (over more years)? All the irrigation and rainfall were effective? what was the efficiency? I think this is important because it was a two year study and rainfall may vary year to year. Therefore, the irrigation level recommended in the conclusions will depend on the year. I will suggest the authors to include ET in the Figure where the rainfall and temperature is provided or include one with ET and irrigation plus rainfall for all the treatments.
I would like to know what is the row spacing used?
Author Response
Dear Editor,
We sincerely thank the reviewers for their comments regarding our manuscript. We have carefully revised the manuscript according to their suggestions. We hope these modifications are acceptable to you. Thank you again for your time and consideration.
1. I think this is a good paper. However, the only problem I have with this paper is that the experiment consisted in applying different irrigation rates to evaluate the response to MC. However, the authors don't mention details about how irrigation was calculated. How they decided when to apply irrigation? How the different irrigation regimes were decided? It seems that the highest irrigation treatment was to apply 6 cm per irrigation with 8 irrigations. The lowest irrigation treatment was to apply an irrigation depth of 3.6 cm in 8 irrigation events. The irrigation was applied in the same dates for all the treatments?
Response: Thank you very much for your comments. The yield and water use efficiency of cotton are closely related to the amount of irrigation water in the whole growth period. High yield cannot be achieved if the amount of irrigation is too small or too large. Therefore, we mainly referred to the practice of farming irrigation in Xinjiang to determine the irrigation regimes (Cai et al., 2002; Liu et al., 2011; Zhang et al., 2015) and the field experiment were reproducible in the region. We also mentioned the basis for the formulation of the irrigation regimes in the experimental design.
The irrigation was started 45 days after emergence of cotton. The total amount of irrigation is 4800 m3 as the conventional irrigation, and based on this value, 90%, 80%, 70%, 60% of conventional irrigation were reduced as W2, W3, W4, W5, respectively. The first irrigation was carried out on squaring stage of cotton. The irrigation was applied in the same dates for all the treatments, and the irrigation time was generally 10-14h (07:30 AM-21:30 PM) due to the difference in irrigation amount.
Cai, H. J.; Shao, G. C.; Zhang, Z. H., Water demand and irrigation scheduling of drip irrigation for cotton under plastic mulch. Journal of Hydraulic Engineering 2002, 11, 119-123.
Liu, M. X.; Yang, J. S.; Li, X. M.; Yu, M.; Wang, J., Effects of irrigation amount and frequency on soil water distribution and water use efficiency in a cotton field under mulched drip irrigation. Chinese Journal of Applied Ecology 2011, 22, 3203-3210.
Zhang, H. L.; Luo, H. H.; Li, L.; Zhang, Y. L.; Zhang, W. F., Characteristics of root and shoot biomass accumulation in high-yield cotton fields with mulch-drip. Cotton Science 2015, 27.
2. What was the evapotranspiration, and how much of that ET was supplied by irrigation? What is the average annual rainfall for that area? Is it variable with time (over more years)? All the irrigation and rainfall were effective? what was the efficiency?
Response: Thank you very much for your comments. Evapotranspiration of cotton during the growth period was calculated according to a hydrologic balance equation (Kang, et al., 2000).
ET = Rainfall + Irrigation + Sg - D - Rf - △W
where ET is evapotranspiration (mm), â–³W is the soil water content change (mm), Sg is the capillary contribution from groundwater table to the crop root zone, D is the downward drainage from the crop root zone, Rf is the surface water runoff. The terrain is flat and the amount of irrigation was controlled as well as the rainfall was very small during the growth period, so Sg, P and Rs can be ignored and they were assumed to be zero.
Kang, S. Z.; Shi, P.; Pan, Y. H.; Liang, Z. S.; Hu, X. T.; Zhang, J., Soil water distribution, uniformity and water-use efficiency under alternate furrow irrigation in arid areas. Irrig. Sci. 2000, 19, 181-190.
Climate in this region was temperate continental with total annual precipitation of 125.0-207.7 mm and annual evapotranspiration of 1425 mm. Rainfall in the region was very small and annual evaporation was much larger than annual rainfall, so the rainfall was almost ineffective. In addition, the mulched drip irrigation was widely used in the region because of drought and water shortage to avoid large amounts of soil water evaporation between rows and reduce deep percolation.
3. I think this is important because it was a two year study and rainfall may vary year to year. Therefore, the irrigation level recommended in the conclusions will depend on the year. I will suggest the authors to include ET in the Figure where the rainfall and temperature is provided or include one with ET and irrigation plus rainfall for all the treatments.
Response: Thank you very much for your comments. We have added the information about ET (mm) during the growth period in the Figure 1.
4. I would like to know what is the row spacing used?
Response: Thank you very much for your comments. The row spacing was managed as 12 cm with a planting density of 18,000 plants ha-1 which is commonly practiced in this region (Hu, et al., 2012).
Hu, Y. Y.; Zhang, Y. L.; Luo, H. H.; Li, W.; Oguchi, R.; Fan, D. Y.; Chow, W. S.; Zhang, W. F., Important photosynthetic contribution from the non-foliar green organs in cotton at the late growth stage. Planta 2012, 235, 325-36.
Reviewer 2 Report
Some minor editing to the manuscript will improve its readability. For example, several mentions of 'assimilates' would be better as 'assimilate' as the singular refers to multiple compounds already. I suggest line 34, page 1 be changed to: 'Although cotton is considered a drought-resistant crop, its productivity can be negatively affected by drought stress.' Line 70: the Latin name for cotton should be italicized. Line 23, 'boll opening (BO) is 122 days' not 'are 122 days'. Line 90 is unclear: is the 2.7 liters per hour flow rate for the 55 cm distance, or per 100 meters, or what? Usually flow rate is shown as a volume per linear distance of dripper line. Line 96: change 'the urea' to 'urea'. Line 129 '105 degrees C for 30' what? minutes? No unit is shown after 30, so add minutes, hours, or whatever the time span was. Table 2: under seed yield column the statistical significance should be ns instead of Ns. Lines 297-298 appear to be a fragment. Line 349 Luxury is misspelled as 'luxary'. Line 386 in the conclusion (and in the abstract) the drip irrigation level of 540-600 m3 hm-2 perhaps should be m3 per ha? Nowhere else do I see the units of m3 hm-2 in the manuscript. The conclusion that WP was improved without reduction in yield... may be overstated.
The authors are to be complimented on the work reported here. The large amount of work to conduct the experiments, to analyze the data, and to produce the manuscript is impressive.
Author Response
Dear Editor,
We sincerely thank the reviewers for their comments regarding our manuscript. We have carefully revised the manuscript according to their suggestions. We hope these modifications are acceptable to you. Thank you again for your time and consideration.
1. Several mentions of 'assimilates' would be better as 'assimilate' as the singular refers to multiple compounds already.
Response: Thank you very much for your comments. We have revised 'assimilates' to 'assimilate' in the text.
2. I suggest line 34, page 1 be changed to: 'Although cotton is considered a drought-resistant crop, its productivity can be negatively affected by drought stress.'
Response: Thank you very much for your comments. We have modified in the text.
3. Line 70: the Latin name for cotton should be italicized.
Response: Thank you very much for your comments. We have modified in the text.
4. Line 73, 'boll opening (BO) is 122 days' not 'are 122 days'.
Response: Thank you very much for your comments. We have modified in the text.
5. Line 90 is unclear: is the 2.7 liters per hour flow rate for the 55 cm distance, or per 100 meters, or what? Usually flow rate is shown as a volume per linear distance of dripper line.
Response: Thank you very much for your comments. We have revised the wrong expression of‘The inner diameter of the drip irrigation line was 2.3 cm with 55cm distance of the dripper and 2.7 L h-1 flow rate’ to ‘The drip irrigation line had an inner diameter of 2.5 cm, a emitter distance of 50 cm, and a flow rate of 2.7 L h-1.
6. Line 96: change 'the urea' to 'urea'.
Response: Thank you very much for your comments. We have modified in the text.
7. Line 129 '105 degrees C for 30' what? minutes? No unit is shown after 30, so add minutes, hours, or whatever the time span was.
Response: Thank you very much for your comments. We were sorry for the wrong description, we have modified to '105 degrees C for 30 minutes'.
8. Table 2: under seed yield column the statistical significance should be ns instead of Ns.
Response: Thank you very much for your comments. We have modified that in the table 2.
9. Lines 297-298 appear to be a fragment.
Response: Thank you very much for your comments. We were sorry that the conjunctions were not used correctly, resulting in fragmentation of the statements. We revised to ‘Hence, irrigation strategies can be used to alter leaf area expansion, the absorption of photosynthetically active radiation and carbohydrate production to increase the potential of photosynthesis, water conservation and consequently yield.
10. Line 349 Luxury is misspelled as 'luxary'.
Response: Thank you very much for your comments. We have modified that in the text.
11. Line 386 in the conclusion (and in the abstract) the drip irrigation level of 540-600 m3 hm-2 perhaps should be m3 per ha? Nowhere else do I see the units of m3 hm-2 in the manuscript.
Response: Thank you very much for your comments. We have modified that in the conclusion and abstract sections.
12. The conclusion that WP was improved without reduction in yield... may be overstated.
Response: Thank you very much for your comments. We have revised the conclusion in a discretionary manner. AS following: In this study, irrigation quota and MC application had significant effects on photosynthetic performance and biomass accumulation, further affecting cotton yield, quality and water productivity.
Round 2
Reviewer 1 Report
none